# A Sustainable Management Model for Cultural Creative Tourism Ecosystems

**Blanca García Henche [1],\*, Erica Salvaj [2,3] and Pedro Cuesta-Valiño [1],\*** 

[1]  Department of Economics and Business Management, University of Alcalá, 28802 Alcalá de Henares, Spain
[2]  School of Business and Economics, Universidad del Desarrollo, Santiago 7610658, Chile; esalvaj@udd.cl
[3]  School of Business, Torcuato Di Tella, Buenos Aires C1428BCW, Argentina
\*   Correspondence: blanca.garcia@uah.es (B.G.H.); pedro.cuesta@uah.es (P.C.-V.)

**Abstract:** This article proposes a novel management model for cultural, creative, and historic tourism cities. The creation of the model is based on previous literature and in the study of Barrio de las Letras, in Madrid, to identify the key components to successfully develop creative tourism ecosystems. The model integrates the literature on city center management and, unlike previous studies, incorporates missing elements, such as the role of small businesses associations and collaboration networks among diverse stakeholders to develop a cultural–historic tourism ecosystem. This model represents a proposal that supports the coexistence of the private and public sector and sustainable governance models that integrate the inhabitants of city centers with the economic activity generated by urban tourism. The model was developed by an analysis of secondary sources, interviews with key informants, and questionnaires of entrepreneurs located in a recently invigorated cultural and historic neighborhood. The contribution of knowledge offered by this paper is the proposition of a management model that can aid town centers to create competitive cultural/creative/historic tourism ecosystems while still preserving the sustainability of their social/commercial fabric. Therefore, the collaboration of cultural organizations, hospitality industry and retail can promote cultural, creative, and sustainable management model of historic urban centers.

**Keywords:** historic city centers; cultural and creative tourism; urban tourism; sustainable governance; social networks; Barrio de las Letras

## 1. Introduction

Cultural and historical tourism is a growing global phenomenon [1]. The World Tourism Organization, in its sectorial study "Panorama 2020", notes that the anticipated growth in tourism is largely a response to the human need for contact, together with the pursuit of cultural authenticity and sustainability [2–7]. Individuals need to explore their emotions in a controlled manner and the search for immediate sensations and affective experience are trends in today's society [8].

Long journeys are expected to increase, but in parallel, short trips to global city destinations will maintain their appeal, especially in Europe, reflecting tourists' deep interest in culture and history [9,10]. Several studies affirm the gradual increase in the influx of visitors to European and Spanish urban centers where the main attraction is their historical heritage [11–14].

The community of Madrid broke its record of tourists in 2017, reaching 12 million visits. According to data provided by Exceltur [15], Madrid is the Spanish community where tourism increased the most in 2017: 10.2 per cent. The latest survey by the public company Madrid Destination (2017) draws a profile of the tourist who comes to the capital, a tourist interested in culture, museums, gastronomy, night life, and Spanish hospitality [16].

Therefore, according to both Madrid Destination and Turespaña [17], the attraction of new market niches has been set as a goal: a cosmopolitan tourist wanting to soak up culture, history, traditions, and uniqueness. It is this offer of culture, tradition, and uniqueness which plays an important role in the different neighborhoods of the capital, such as in the Barrio de las Letras in Madrid. Currently, to Madrid, these tourists provide growth through the revitalization of neighborhoods, restoration of buildings, boosting trade, and targeting new market segments, among other things.

Barrio de Las Letras represents an example of revitalization of historic districts in which those new visitors are increasingly interested: they seek to move away from the crowds and prefer the alternative historical, cultural, culinary, artistic, or social activities within a particular space, along with a social system that offers a unique identity.

Noting such trends of revitalization of historic town centers and neighborhoods, scholars involved in community based tourism research since Murphy's book [18], "Tourism a community approach" (1985), have called for studies that develop appropriate management models for cultural and historic tourism that at the same time respect authenticity and sustainability in such city centers [19–27]. The present research responds to this call by reconceptualizing and proposing a management model that can aid town centers looking for competitive cultural and historic tourism ecosystems [28,29], while still preserving the sustainability of their local social and commercial fabric. This work is thus relevant in several respects. First, historic urban centers have extraordinary growth potential, considering the vast increase of cultural and historic tourism in recent years [30,31]. Secondly, old city centers are more complex and challenging than commercial town centers; not only for the presence of small retailers, but social, historical, and cultural entrepreneurial activities are integral elements of these districts. Leadership and management in such environments emerge from the local actors and is about serving a diverse community with different interests rather than leading a company with profit-oriented goals [32–34], so novel models must acknowledge this distinctive reality. Thirdly, large populations live in these districts, together with the presence of cultural and historic tourism, creating a prominent challenge in terms of developing sustainable models that integrate local communities with economic activities, without expelling them from their natural environment [27,35,36].

To build the present model, this study considered research into town center management (TCM) [37–39], cultural and historic districts [24,40–42], social network theory [36,43–45], and the in-depth case study of Barrio las Letras, in Madrid [46]. By combining these sources, several components were identified that are critical to managing successfully cultural and historic tourism ecosystems. The model integrates components, such as the relevance of a geographically limited space, the role of business associations and social relations, and the potential for collaboration among all the stakeholders involved.

This study is focused on reconceptualizing the management models for creative city centers with historical and cultural identities.

Two main lines of research detail models of urban management [39,47,48]. Cohen and Applebaum (1960) introduce the concept of business improvement districts (BIDs), known as the American model [49]. Their evaluations of store locations mainly refer to sales and profit potential, such that the model delineates and identifies areas with greater sales capacity. These BIDs generally overlap with town centers, so in this view, property owners should be charged taxes, with these revenues used to improve the quality of the public services and make the commercial areas more attractive [37,39,41,50]. For this study setting, this line of research is limited though, because it only considers retail as a productive activity and is focused on profit oriented goals ignoring the interests of other stakeholders as Valente, Dredge, and Lohmann (2015) pointed out [32].

As town centers gradually lost their commercial appeal [25], public and private actors have implemented different practices in recent decades to regenerate urban centers. With a broader perspective, Kotler, Haider, and Rein (1993) offer the concept of TCM, also known as the European model [51]. They seek a more complete and comprehensive view, involving the management and promotion of both public and private areas within town centers, for the benefit of all

stakeholders. This definition addresses concerns beyond retail, including sustainable development, local administrations, community engagement, services, and regeneration [25].

To address the variety of issues in the stewardship of cultural and historical resources and the impact of tourism on host communities [18], four models of TCM have been identified based on the sectorial affiliation and the extent of formality: the private-informal (e.g., trader association led schemes), the public-informal (voluntary/community led schemes), the private-formal (e.g., limited company, chamber of commerce led schemes) and the public-formal [38]. The TCM models thus overcome some of the limitations of the BID model, yet they remain focused on the "commercial city," without considering other levels that overlap in town centers, such as the "cultural and historic city" [1,26,52]. These levels usually intersect in the same geographically delimited area. Cultural districts have been defined as systems of interdependent entities—including public and private institutions, businesses, entrepreneurs, individuals, and local communities—situated within a limited geographical area, aimed at achieving sustained value creation and driven by the unifying role of culture and creativity [53,54].

Cultural districts evolve from a mixture of top-down planned and emergent activities involving a large set of stakeholders belonging to different value chains. Within such a context, local culture, museums, and traditions are fundamental ingredients to create the idiosyncrasy of the culture and tourism products of a district [40]. Focusing on these ingredients, creative tourism bonds people to places, promoting tourist immersion into the local culture and the active participation in cultural and creative activities [55] and rediscovering regions' identities—something that cities could use to create sustainable offers for tourists.

Networks of stakeholders are key instruments for building collaboration, as well as for shaping and developing a tourism cluster [44,45]. Several empirical studies identify social networks as key for developing collaborations and creating benefits, in terms of the development and sustainability of a tourist destination [56–58].

Previous studies on collaboration networks in the tourism industry based in countries such as Australia [59], Italy [60], or Spain and Chile [61] support these arguments. Additionally, in the United Kingdom, Novelli, Schmitz, and Spencer (2006) analyze how social networks help improve the performance and sustainability of small tourism enterprises [62], and Czakon and Czernek (2016) analyze the Polish tourism industry and highlight the importance of establishing trusting relationships to generate cooperation in the networks [63]. In the case of cultural and historical tourism, it is argued that it is necessary to include into the picture a tourism network approach [43] to capture the mechanism of collaboration to help develop cultural and historical tourism.

These studies show how collaboration networks improve the competitiveness of the sector, which is now more necessary than ever [64]. Cooperation and networks are a true multiplier of opportunities in the tourism sector [65] because they help to enhance the transfer of knowledge and experiences and innovation [36].

On the other hand, in contexts of technological change and turbulence such as the one we are experiencing, networks are key to developing trust, and transmitting information and knowledge that facilitate adaptation to new technologies and knowledge [66]. Collaboration is also a crisis survival strategy, as it involves coordination, communication, cooperation, and knowledge transfer [35].

Accordingly, this article is organized as follows: Firstly, the study presents a review of the literature on TCM, cultural and historical districts and social networks in tourism and describes a novel model of management for cultural and historic tourism environments. Secondly, the research explains the methodology. Then, a precise description of the experimental results is provided.

Finally, the study discusses the results describing Barrio de Las Letras and its cultural and leisure services and products and, finally, this research presents a management and organizational model based on the case of Barrio de Las Letras in Madrid. This study concludes with the theoretical and practical implications and suggestions for future research directions.

This article sheds light on the importance of reconceptualizing the management models for creative city centers with historical and cultural identities (such as Berlin's Kreuzberg district, Amsterdam Noord

in Amsterdam, Norrebro in Copenhagen, Gracia in Barcelona, East End in London, Sodermalm in Stockholm, Le Marais in Paris, Capucins in Bourdeaux, Palermo in Buenos Aires, Barrio Italia in Santiago de Chile, Chūō-ku in Osaka (Japan), Banglampoo in Bangkok in Thailand, or Xintiandi and the French Concession area in Shanghai). It contributes to a better and more detailed understanding of networking in creative and cultural cities, integrating, as key stakeholders, not just commercial stores and entrepreneurs, but also cultural institutions, such as museums and scientific, literary, and artistic centers.

## 2. Methodology

The model that is presented was motivated by an analysis of secondary sources, interviews to key respondents and 187 questionnaires of entrepreneurs and owners of stores located in a recently invigorated cultural and historical neighborhood, Barrio de las Letras in Madrid. Inductive analysis of the sources revealed the importance of the elements included in the model and suggested initial ideas about how it might promote the successful development of historical and cultural tourism.

However, given some limitations of the data, this study engaged in deductive theory-building to develop a model encompassing both insights from prior research and those that first caught our attention through the interviews and surveys. Our aim was to build and test the proposed model, but we recognize that it can be further refined and empirically validated in future studies.

First, the model was tested by reviewing secondary sources of information about this urban area. Secondly, primary sources are solicited to obtain first-hand information about the characteristics of its cultural, commercial, and historic offers. The open-ended interviews included key respondents from Barrio de Las Letras such as:

- The president and founder of the Asociación de Comerciantes del Barrio de las Letras (The Association), a 66 year-old man, who has been an entrepreneur in the neighborhood for 35 years and owns one of the oldest shops. The topics addressed in this interview were trade in the neighborhood, local entrepreneur's profiles, historical changes in neighborhood stores, goals, and commercial actions of the association.
- The general manager of The Association, a 49 year-old woman with a university degree. The subjects addressed were the goals, activities and commercial actions of the association, the responsibilities, and main managerial challenges of her position.
- Two additional founders of The Association: both male entrepreneurs, 45 and 48 years old, owners of a sports specialized store. The topics addressed with them were the reasons for the creation of the association, historical changes in the profiles of the entrepreneurs in the neighborhood, and the activities of The Association.

These interviews revealed the importance of the development of cultural tourism, the social dynamics, and the processes that take place in Barrio de las Letras. Guided by the insights provided by the interviews and the information collected from secondary sources, this study developed a 2-part questionnaire. The first part gathered information about the characteristics of the entrepreneurs and small retailers and the commercial actions they conducted. The second part, using the approach developed by Cross and Parker (2004), sought to understand the articulation of the small shops and entrepreneurs' network [67] and the extent to which collaboration existed among them, small shops, and cultural institutions of Barrio de Las Letras.

The questionnaire was administered to the members of the Association during November–December 2015, in the district of Barrio de Las Letras. 187 completed questionnaires were collected from the target population of small and medium retail, lodging, services, hospitality, cultural, and historic organizations. The technical details of the data collection are in Table 1. Social network analysis was applied [68] to calculate centrality and structural measures, and Gephi software was used to generate graphs and estimate individual network measures, such as degree and betweenness centrality, as well as structural measures such as cohesion.

**Table 1.** Descriptive information about the sample.

| Universe | Commercial and Cultural Organizations. Barrio de las Letras |
|---|---|
| Geographical Area | Barrio de las Letras—Madrid |
| Sample | 187 valid surveys (of the 301 associates businesses) |
| Sampling Procedure | Convenience sampling, setting a confidence level of 95% and corresponding to a level of significance of 5% |
| Sample Error | According to the sample size used, the maximum permissible error (for an estimation of proportions) is assumed to be +/- 4% under conditions of maximum uncertainty ($p = q = 50\%$) |
| Technique of collection of the information | Managed personal survey with structured questionnaire |
| Period of collection of information | January–July 2016 |
| Information processing | Univariate and descriptive bivariate analysis with Dyane, SPSS 12.0. and Excel. Social network analysis with Gephi |

Source: Own elaboration.

## 3. Results

### 3.1. Management Model for City Centers with Historical and Cultural Identities: The Case of Barrio de las LetrasSubsection

Barrio de las Letras is in the heart of Madrid and is a member of the Confederation of Historical Centers. Its cultural atmosphere is generated mostly by commercial and cultural ventures. As part of the historical center of Madrid, it is very attractive for commercial, social, and cultural activities. In recent years cultural and historic tourism has developed successfully in the area. Visitors to historic and cultural heritage has increased significantly), while the number of hotels and restaurants has grown exponentially ().

By integrating TCM literature with cultural tourism research and a social network approach the study seeks to build a management model for cultural and historic tourism ecosystems in town centers.

We found that a private–informal management model fits with the creation of experience products, such as cultural and historic tourism, because they demand the collaboration and involvement of several stakeholders, such as residents and tourists in the design of such experiences.

In Figure 1, the management model for historic city centers is presented (Figure 1) that help to develop cultural tourism ecosystems such as Barrio de Las Letras (Figure 2). This model is suitable for districts that include the commercial, cultural, and historic city [1], and it includes: (1) the delimited urban space, which is the historic city center; (2) the cultural, commercial, and service supply, including specialized commerce, historical and cultural heritage, and tourism and cultural services; and (3) the organizational and management model, which considers an entrepreneurs' associations with a manager [61,69] and the informal collaboration networks among different stakeholders.

The three components in this system must communicate each other and align consistently to create an effective cultural and historic tourism ecosystem. In this process the role of The Association that emerges from the local actors is key and it keeps them aligned, and the informal social networks among all the actors help to share knowledge and information and to develop creative and interesting services and activities for tourists.

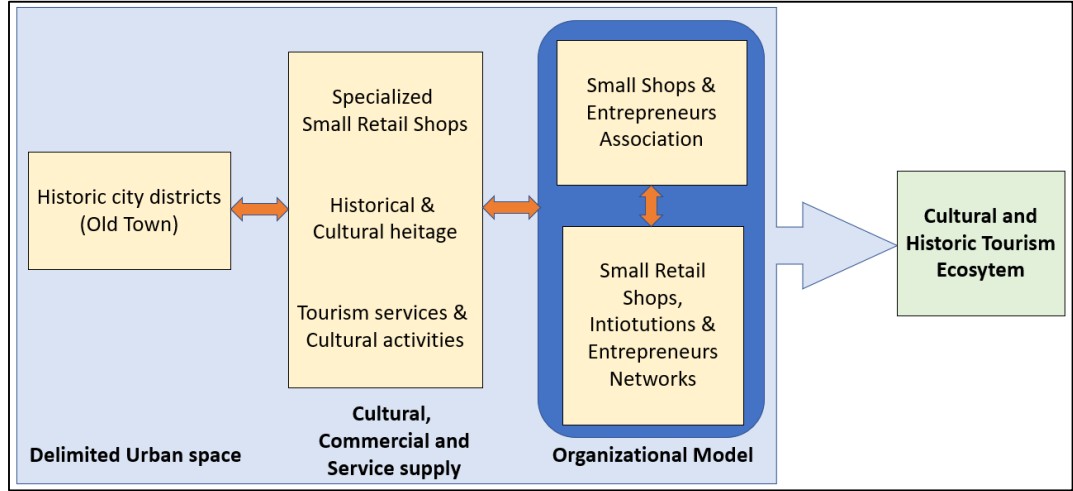

**Figure 1.** A general model of governance for a cultural and historic tourism ecosystem. Source: own elaboration.

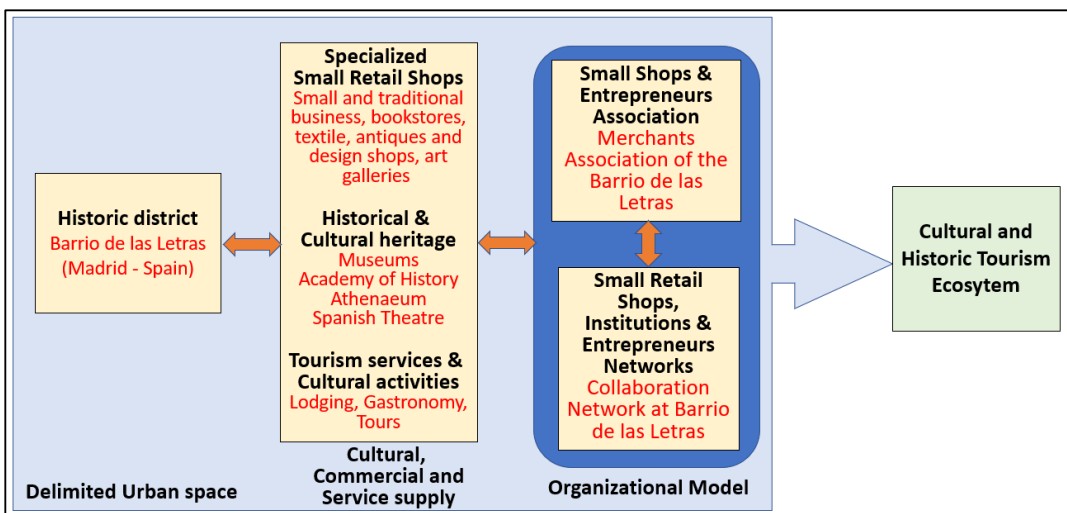

**Figure 2.** Management model for a cultural and historic tourism ecosystem in Barrio de las Letras. Source: own elaboration.

### 3.2. Delimited Urban Space in Barrio de las Letras

Historic town centers often emerge in polycentric world cities [42], constituting the spaces that give towns their special character and make them unique [70]. They are limited by a set of streets [40] that surround the most symbolic and appreciated monuments or distinctive places of these cities [71]. Historic neighborhoods that have been ignored by prior research have acquired greater visibility, emerging as poles of tourist attraction or "micro-destinations". Barrio de Las Letras represent a case of these revitalized historic districts, interesting for a new type of visitor who seeks to get away from the crowd and prefers alternative historic, cultural, culinary, artistic, or social activities within a particular space, along with a social system that offers a unique identity. This district is limited by four streets: Atocha, Paseo del Prado, Carrera de San Jerónimo, and Carretas (see Figures 3 and 4). Its remarkable historical resources include the Cervantes House and the Ateneo, as well as some of the most important museums of the city (Prado, Reina Sofía, Thyssen Bornemisza).

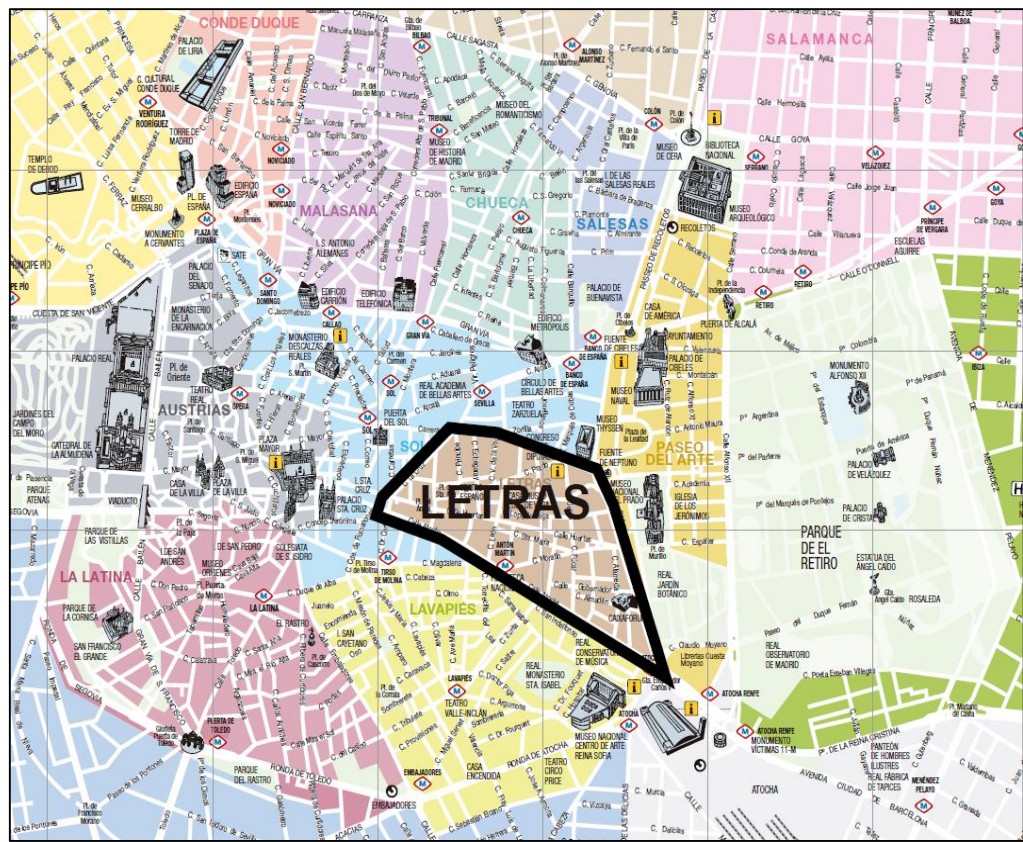

**Figure 3.** Map or tourist areas in Madrid and Barrio de Las Letras's location. Source: esmadrid.com [72].

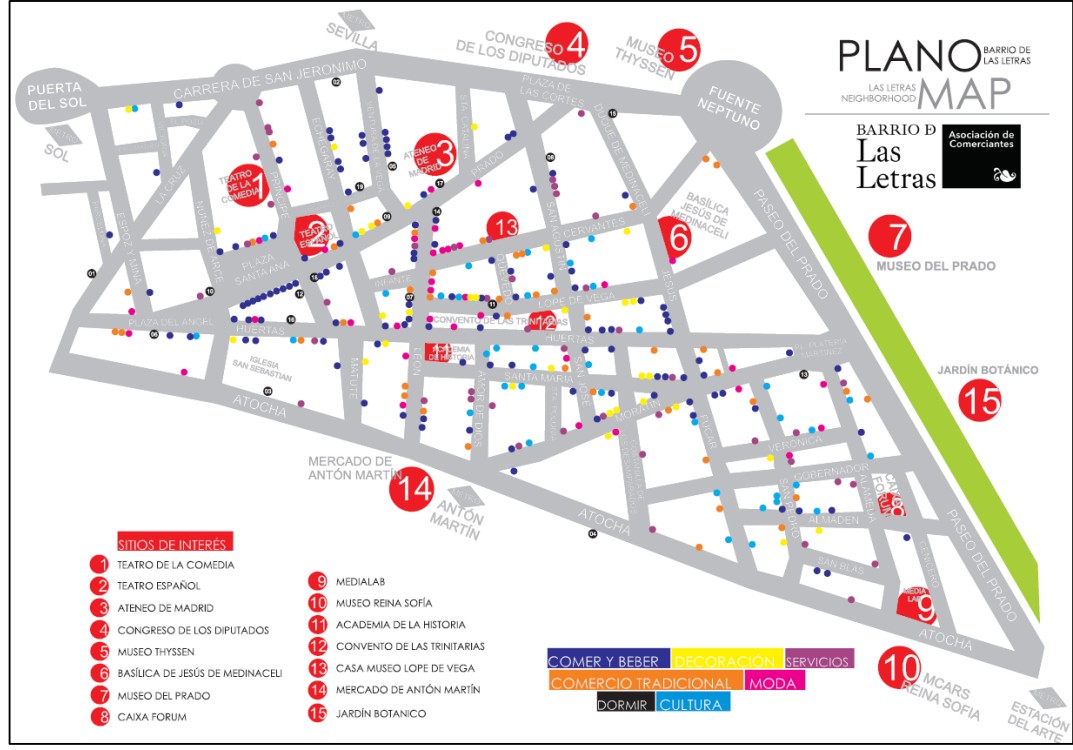

**Figure 4.** Map of Barrio de las Letras. Source: The Association Barrio de Las Letras [73].

*3.3. Cultural, Commercial, and Service Supply in Barrio de las Letras*

Barrio de las Letras is dynamic, well connected, and marked by a variety of cultural and leisure services and products, organized into three components.

- Specialized small retail. The influence of retail activity on the transformation of global cities is evident [69]; local and specialized shops have strategic value for their capacity to create unique environments that attract tourists. In some cities, the shopping streets are cultural objects, because of their specialized products. Specialized retail also might be associated with the historical and cultural identity of the district, such as the case of the "Centenary Stores of Madrid"; shops that are part of the living history of the city, protected and promoted in a Madrid tourism-sponsored guide, of which many are located in Barrio de las Letras. According to the data, the small retail segment consists of small commercial shops (95 respondents), with 35% dedicated to gastronomic services (67 respondents), 10% to diverse services (19 respondents), and 5% to lodging (9 respondents). The commercial establishments that create the identity of the neighborhood are mostly associated with history, art, and culture. The size of these businesses is small, with just an average of seven employees, though many businesses have just two or three employees. In our sample, 50% of the respondents to the questionnaire were the owners of the establishments, and the other 50% were employees who run the small businesses. In addition, 65.78% of the businesses are managed by people with university degrees, and 23.53% have secondary level. The deviation (16.76) stems from the hospitality and catering firms, which require more employees (around 20).

Table 2 details establishments by activity. Various specialized shops reference the local history and culture, such as antiquarian shops (almost 10%), book shops (over 6%), art galleries (almost 16%), and fashion and jewelry shops (nearly 25%) (several establishments host workshops at the same location where they sell their products). Moreover, restaurants have grown notably in recent years, generating an increasing number of jobs. In Barrio de las Letras, 35% of establishments belong to this sector. A total of nine lodging establishments were also surveyed: 66.67% hotels, 11.11% hostels, and 22% pensions [46].

- Historic and cultural heritage. The local legacy is not necessarily associated with traditional tourist circuits, which highlight great monuments or visual icons that attract millions of visitors each year. Historic and cultural heritage instead constitute a strategic asset that is fundamental for developing cultural tourism and has a significant impact on destination brand equity [74–76] and includes tangible (e.g., museums, historical monuments, centenary stores) and intangible (e.g., routes, fairs, exhibitions, congresses, festivals) resources. "Heritage tourism is more "place-based" in that it creates a sense of place embedded in the local landscape, architecture, people, artefacts, traditions while cultural tourism is broadly concerned with the same types of experiences as heritage tourism, but at the same time less concerned with place" [77]. Intangible cultural heritage is a key means to differentiate cultural and historic destinations that benefit not only the socio-cultural life of visitors who are looking for unique experiences but also the inhabitants who live there [78]. Barrio de las Letras is a unique combination of heritage, cultural, and literary tourism. Literary tourism is associated with "places celebrated for literary depictions and/or connections with literary figures" [77]. Barrio de las Letras includes first-class historical and cultural offerings, such as the Prado Museum, Reina Sofía Arts Center, Caixa Forum, Lope de Vega and Cervantes House, and Ateneo de Madrid. With this cultural identity, the neighborhood is known for its unique "art strip". Recent promotions of related cultural activities by these institutions have greatly increased the number of visitors (Table 3).

- Tourism services and cultural activities. Cities exposed to mass urban tourism need to find ways to decongest their traditional historic areas and promote new districts. Many and varied stakeholders participate in the reinvention of cultural and historic tourism, creating a mixture of top-down

planned and emergent activities that seek to take advantage of the specialized retail offerings of the district, as well as the historical and cultural heritage already available there [40,80,81].

**Table 2.** Disaggregated types of business activity in Barrio de las Letras.

| Shops | Frequency (95) | Percentage |
|---|---|---|
| Antiquities | 9 | 9.47% |
| Food | 8 | 8.42% |
| Art | 15 | 15.79% |
| Flowers and plants | 1 | 1.05% |
| House and decoration | 6 | 6.32% |
| Musical instruments | 1 | 1.05% |
| Book shops | 6 | 6.32% |
| Fashion and Apparel | 20 | 21.05% |
| Health | 2 | 2.11 |
| Jewelry and costume | 3 | 3.16% |
| Printing and graphic arts | 1 | 1.05% |
| Other commercial activities | 12 | 12.63% |
| Repair and sale of motorcycles | 1 | 1.05% |
| Restaurants/Catering | Frequency (67) | Percentage |
| Restaurants and catering | 67 | 35% |
| Lodging | Frequency (9) | Percentage |
| Restaurants and catering | 67 | 35% |
| Hotels | 6 | 66.67% |
| Hostels | 1 | 11.11% |
| Others | 2 | 22.22% |
| Services | Frequency (19) | Percentage |
| Beauty | 2 | 10.53% |
| Teaching | 1 | 5.26% |
| Gym and Spa | 1 | 5.26% |
| Real State | 3 | 15.79% |
| Health and Sanitary centers | 1 | 5.26% |
| Leisure and Free time | 1 | 5.26% |
| Other services | 10 | 52.63% |
| Beauty | 2 | 10.53% |

Source: Own elaboration.

The revitalization of the neighborhood is clearly supported by the growth of tourism. The increase in the number of hotels is an unmistakable sign of the vitality of the industry. Figure 3 reveals the increasing number of hotels that have opened since 2005 (Figure 5). This figure does not consider pension accommodations, or the apartments included on platforms such as Airbnb.

**Table 3.** Visitors to museums (thousands).

| Museum | 2010 | 2011 | 2012 | 2013 | 2014 | 2015 | 2016 | 2017 | 2018 |
|---|---|---|---|---|---|---|---|---|---|
| Prado | 2.732 | 2.912 | 3.157 | 2.516 | 2.754 | 2.697 | 3.034 | 2.824 | 2859 |
| Reina Sofía | 2.414 | 2.706 | 2.572 | 3.185 | 2.677 | 3.257 | 3.745 | 3.896 | 1694 |
| Lope de Vega | 21 | 22 | 22 | 42 | 64 | 84 | 106 | 90.2 | 151 |
| Caixa Forum | - | 1.000 | 904 | 764 | 768 | 568 | 568 | 623 | 947 |
| MADRID | 9.521 | 11.513 | 11.835 | 11.653 | 12.445 | 13.083 | 13.723 | 10.756 | 16.448 |

Source: www.Madrid.es (2020) [79].

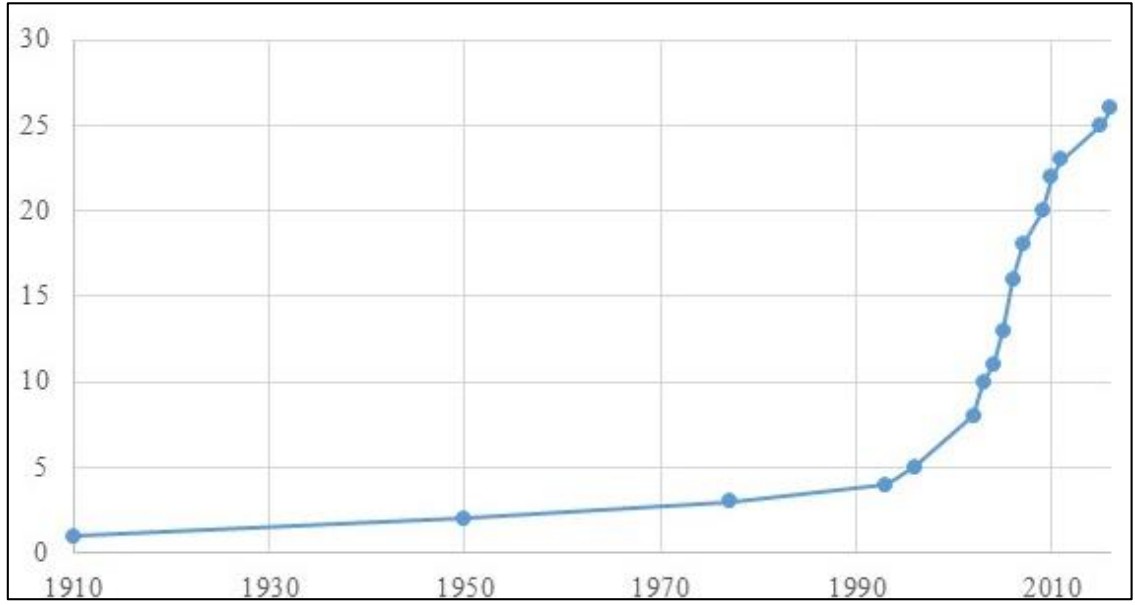

**Figure 5.** Evolution of Hotels (total number of hotels in Barrio de las Letras). Source: The Association Barrio de Las Letras [73].

Cultural events serve as promoters of tourism and help to develop destination image [75,82]. Many stakeholders, including The Association and its members, participate in devising a good mixture of activities and experiences to take advantage of the commercial, historical, and cultural heritage that already exists in Barrio de las Letras. These activities are closely aligned with the identity of the area, representing experiential marketing and cultural activities that seek to promote the "Barrio de Las Letras brand". The actions listed in Table 4 are organized by and rely on the participation of small retail shops and entrepreneurs, which produces an enjoyable atmosphere appreciated by tourists. For example, the Frog Market (El Mercado de las Ranas in which shops of the district take to the street with their commercial, cultural, gastronomic, and artistic offers on the first Saturday of each month) and DecorAcción (organized by The Association, Nuevo Estilo magazine, and the city of Madrid, it encourages design and interior design onto the streets) evoke the greatest participation, media coverage, and number of tourists. Entrepreneurs' involvement in the monthly Frog Market has increased continuously since it began. In 2012, 58.52% of associates (68) participated in the event; in 2015, 116 did. The Association also estimates that 220,000 attendees participated in DecorAcción in 2016 and 200,000 in 2015.

**Table 4.** Participation of The Association members in cultural and commercial actions in 2016.

| Commercial Actions | Advertising Poster | Participation of Members of the Association |
|---|---|---|
| DecorAcción |  | 112 (59%) |
| The Frog Market |  | 116 (62%) |
| The Night of Books |  | 25 (13%) |
| Black Friday |  | 65 (34%) |

Source: Own elaboration and The Association Barrio de Las Letras [73].

### 3.4. Management and Organizational Model

To revitalize historical ciy areas local associations of small retail shops and entrepreneurs and collaboration and social networks are necessary.

- Local associations of small retail shops and entrepreneurs. Associations have a leading role in revitalizing historical city areas [38,83]. Their members tend to be small businesses, looking for collaboration, the protection of their interests, and the creation of a reference group. Associations help coordinate entrepreneurs and small shops; identify new business and tourism opportunities [84]; and fulfill control functions, in that they devise ways to protect the identity of the district and differentiate it from traditional malls in suburbs [85]. Moreover, they often

plan and implement changes to the commercial structures of historic districts, to enhance their attractiveness and visibility. Many studies accordingly demonstrate the relevance of interactions between public administrations and the association, to manage the commercial activity of urban areas jointly [37,84].

The management model of Barrio de Las Letras features an association (founded in 2005) with a board, comprised of members chosen democratically and a manager who reports to them, along with an informal collaboration network among entrepreneurs, retailers, and cultural institutions. The manager facilitates the alignment between the association and the informal network. The general mission is to provide business advice to members and develop Barrio de las Letras as a brand and micro-destination within Madrid.

The association pursues the renaissance of the neighborhood on three working axes: trade, culture, and tourism. First, the rich variety of establishments provides a diverse range of products and services, spanning the original bookstores in Madrid to avant garde art galleries, fashion houses, and antiquaries. Second, culture is present in every stone, with dozens of commemorative plaques identifying historical events and persons who lived there. Third, tourism is the main engine of the neighborhood, and every year millions of people visit to enjoy its rich leisure and gastronomy offerings [46].

The strategy of the association manager and directors stemmed from their belief that tourists consume in a multidimensional way, so the historical district must offer a diverse range of attractions, products, and services. The association, together with the public administration, works on increasing security in the area, cleaning services, maintaining buildings, and encouraging pedestrian traffic. These efforts differentiate the area from other important commercial streets.

The Association receives support from small retailers, entrepreneurs, and institutions, such that it began with 30 members but today has more than 300. Cultural institutions include the Royal Academy of History, Lope de Vega House-Museum, Spanish Theater, Caixa Forum, Monastery of Trinitarias, Medialab, and Ateneo. Of these cultural referents in Madrid, several of them related to literature provide the literary air to the neighborhood which its name refers as a basis for the overall experience offered to tourists.

- Collaboration and social networks. Tourism is a critical industry, due to its vast potential to develop and improve regional competitiveness [65,86,87]. It is difficult to be competitive if regional stakeholders (public and private) are not involved, aligned, and collaborating [88]. Collaboration is essential to generate synergies and combine resources, which can then initiate virtuous cycles [40,86,89,90].

Social networks (personal relationships and interactions) often help the stakeholders look for joint opportunities, resolve problems both individually and collaboratively, and achieve sustainable relationships that define the system in which they are embedded. Networks are key instruments for building collaboration, as well as for shaping tourist destination [43,45]. They offer direct utility to diverse stakeholders in the tourist industry, creating knowledge [57,62].

At Barrio de las Letras, small entrepreneurs, lodging facilities, gastronomy providers, commercial shops, cultural institutions constitute an informal network, which constitutes a key management mechanism necessary to create a successful cultural and historic tourism ecosystem. This collaborative network facilitates quick and effective coordination to generate the commercial actions listed in Table 4. The social interactions thus produced bring all the components of the neighborhood together, increasing the differentiation and visibility of the microdestination brand for Barrio de las Letras and making the district more attractive for urban tourists. Finally, the network enables communication, information sharing, and quick changes to provide high-quality services and adapt to the shifting needs of tourists. Therefore, we find an inclusive informal network of collaboration among entrepreneurs, cultural institutions, and shops (Figure 6). The links we depict in this network reflect the answers we obtained from Part B of the questionnaire. The extent of the ties represents the strength of the relations

among the shops. As Figure 4 shows, the 187 survey respondents mentioned 364 organizations with which they collaborate. Of those, 321 (88.2%) integrate a main component (or subnetwork with the largest number of connected nodes). Therefore, interrelation and collaboration are critical for the success of Barrio de las Letras as a cultural and historical tourism cluster.

The size of the nodes in Figure 6 denote the most active entrepreneurs in terms of collaborating and asking for collaboration, which are important for the creation of a single, interconnected ecosystem. If these entrepreneurs were removed, the network would be damaged and become fragmented. Therefore, it is important to acknowledge the social capital that these entrepreneurs create, not only for their own benefit but for the overall cultural and historic tourism ecosystem. They ensure that the group remains integrated and aligned, and they facilitate the dissemination of information, knowledge, and new ideas that later can be discussed and implemented by various actors. We used the color of the nodes to define the sector. Restaurants and taverns are the most active networkers (7 of the top 20), followed by some hotels (2 in the top 20), food stores (2 of the top 20), pharmacies (2 of the top 20), and silver sellers (2 of the top 20). See Table 5 on the most central actors of the network. Although museums and cultural organizations are integrated, they do not appear active in terms of generating network links.

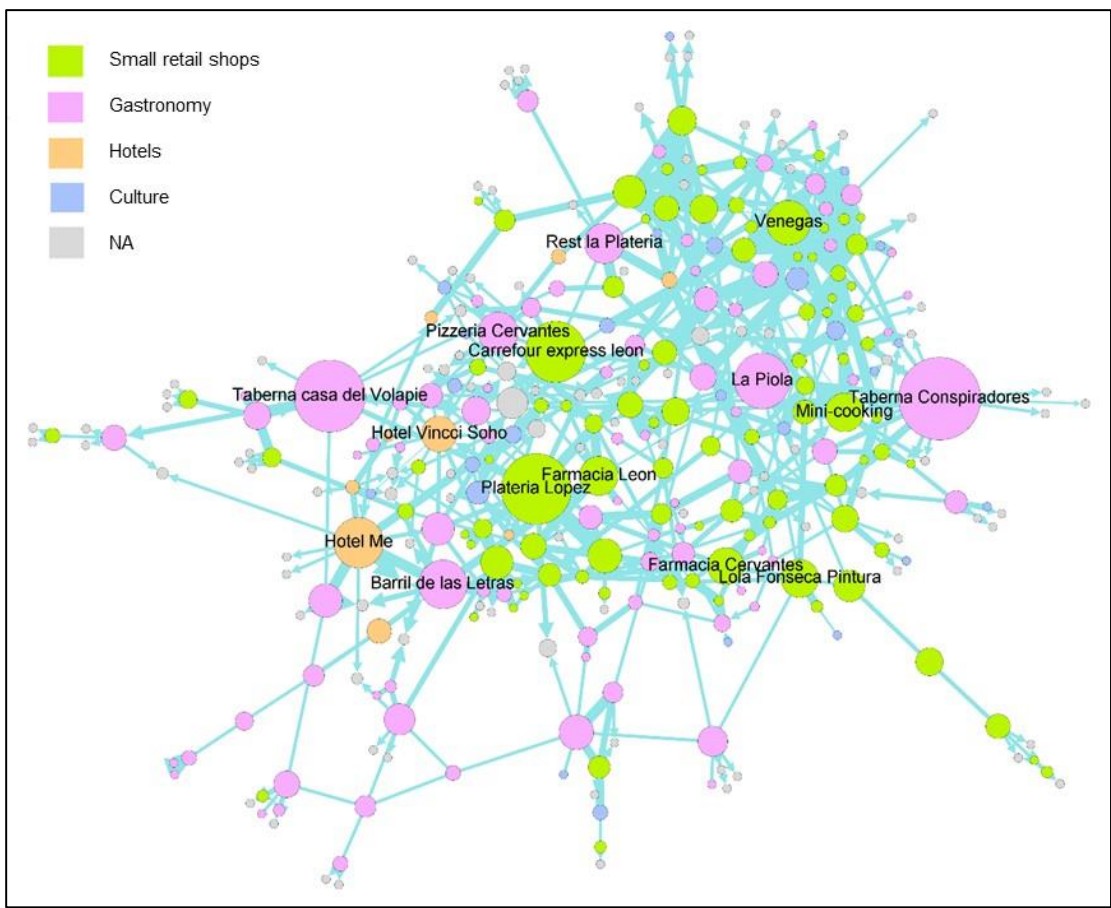

**Figure 6.** Collaboration network among small commerce, entrepreneurs, and cultural organizations. Source: own elaboration.

**Table 5.** Centrality rankings in Barrio de las Letras.

| Betweenness | Name | Degree | Name |
|---|---|---|---|
| 0.11 | Taberna Conspiradores | 19 | Taberna Conspiradores |
| 0.10 | Taberna Casa Del Volapie | 19 | Venegas |
| 0.09 | Platería López | 19 | Érase Una Vez. |
| 0.08 | Carredour Express Leon | 17 | Valyrium |
| 0.07 | Restaurante A'cañada | 13 | Taberna Casa Del Volapie |
| 0.06 | Hotel Me | 13 | Platería López |
| 0.06 | El Barril De Las Letras | 13 | Hotel Me |
| 0.05 | Venegas | 13 | Ginger & Velvet |
| 0.05 | Pizzería Cervantes | 13 | Miseria |
| 0.05 | La Platería | 12 | Hotel Vincci Soho |
| 0.05 | Farmacia León | 12 | Mazarias |
| 0.05 | Mini-Cooking | 12 | Motteau |
| 0.04 | Lola Fonseca Pintura En Seda | 12 | Il Guacciaro |
| 0.04 | Farmacia Cervantes | 11 | Losada |
| 0.04 | Hotel Vincci Soho | 11 | Droguería Pefu Castillo |
| 0.04 | Restaurante Matute | 10 | Carrefour Express Leon |
| 0.04 | Olarra | 10 | Pizzería Cervantes |
| 0.04 | Naturbier | 10 | Mini-Cooking |
| 0.04 | Librería Del Prado | 10 | Olarra |
| 0.04 | Ginger & Velvet | 10 | Bodegas Trigo |
| 0.04 | Restaurante Matute | 10 | Carrefour Express Leon |
| 0.04 | Olarra | 10 | Pizzería Cervantes |
| 0.04 | Naturbier | 10 | Mini-Cooking |
| 0.04 | Librería Del Prado | 10 | Olarra |
| 0.04 | Ginger & Velvet | 10 | Bodegas Trigo |

Source: own elaboration.

As a complement to Figure 6, Table 5 lists the central actors, according to two centrality measures. Betweenness centrality captures the intermediation level of a node, such that establishments with higher betweenness centrality have more control over the network, because more information passes through them. Degree centrality is the number of links or connections of a node, so organizations with a high degree of centrality are more dynamic in creating links.

The density of the network (i.e., number of links to all possible links in the network) is 1.1%, the diameter (i.e., number of steps between the most distant nodes in the network) is 13, and the average local degree (i.e., average number of links) is four. Together, these measures indicate that we are dealing with a network that integrates a large percentage of entrepreneurs in the Barrio de las Letras (88%) but also remains quite dispersed. The density of the network facilitates some synergy among the different sectors, but the association of these diverse stakeholders is not as strong as their physical proximity might suggest. We also uncover varying degrees of compatibility between tourism and other urban functions. The relatively low level of density implies that there are many opportunities to generate more cohesion and collaboration. As Burt (1995) might argue, the network has several structural holes, which means there is plenty of room for collaboration and innovation [91].

The collaboration of the network mainly includes members of the association. Figure 7 is similar to Figure 6, except that we use the color of the nodes to show who are associated (blue) and who are not (orange). As this figure shows, the businesses that are part of the association not only establish collaborative ties with other members but also link to shops that are not part of the association.

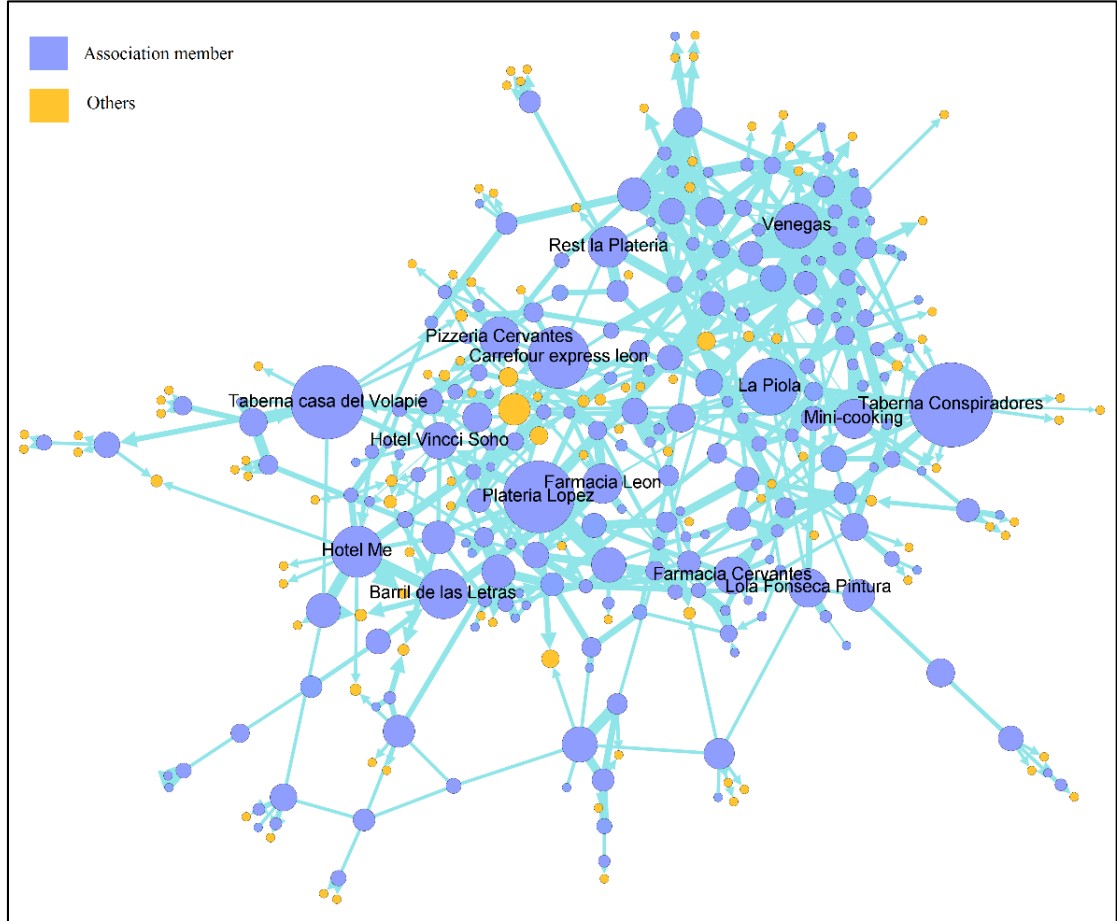

**Figure 7.** Collaboration network among associates and non-associates. Source: own elaboration.

## 4. Discussion and Conclusions

### 4.1. Theoretical Implications

This work captures the changing vision regarding urban centers. The idea of the city has been transformed over the years, from a pragmatic, efficient and productive view to a space that can also generate urban and creative experiences [92].

Despite their increasing popularity, previous research on historical and cultural tourism in urban and large spaces is scarce. This study sheds light on this type of tourism in a global city such as Madrid, and more specifically in Barrio de las Letras. The aim is to contribute to our understanding of how historical districts might reduce some of the costs and risks of urban planning and development [1].

This research makes visible, describes, and analyzes the components that define a creative and cultural neighborhood together with the development of a model of regeneration of urban and historical spaces as a strategy of the capitalization of unique historical neighborhoods.

This study also adds to social network theory in the tourism industry, by exploring how network collaboration facilitates knowledge and experience creation. It shows the informal links among different stakeholders of a cultural district and their collaboration, as well as how these links facilitate the creation of services and experiences that lead to the development of a cultural and historic tourist ecosystem [43,56,93,94].

### 4.2. Managerial Implications

This work contributes to research in management of tourism in urban centers, by integrating TCM, historical and cultural districts literature, and social network research in the tourism industry. In line

with urban regeneration models [69,95], this study develops a model for managing cultural and historic districts in global cities, emphasizing the role of local stakeholders and their self-organizing capability in formal institutions such as associations, as well as in informal institutions such as collaboration networks. They function to develop and manage historical and cultural districts, even without much participation of the public sector, and can create a microdestination brand, such as Barrio de las Letras.

The model presented in this paper produced greater capacity and positive results in Barrio de las Letras. We identify some proxies to measure this success. First, the collaborative and organized work between The Association and the local entrepreneurs to attract the attention of the local authorities. The official web of Tourism of Madrid recently included Barrio de Las Letras in its "neighborhoods" section. Secondly, the most important travelers' guides, such as Guía Total Urban, Michelin Weekend, Guía Vintage, The Monocle, or Trip Advisor, now include and recommend Barrio de las Letras as an interesting location to visit. Thirdly, there was an increasing number of visitors to the historic and cultural heritage (see Table 3) and to the events organized by The Association, for example DecorAccion got 200,000 visitors in 2015 and 234,000 in 2017. Finally, the number of new hotels and restaurants has grown exponentially (see Table 2 and Figure 3).

Furthermore, this study has implications for entrepreneurs and policy makers. First, cultural and historic tourism is a growing trend. New groups of tourists want to be authentically engaged with the territories they visit [96–98] and are looking for unique experiences and activities, eager to visit renewed and creative historic city centers. This movement can facilitate the renaissance of old districts, by creating jobs and new business opportunities [99] for urban entrepreneurs. However, these neighborhoods are also at risk of losing their traditional residents and businesses; gentrification clearly has threatened destinations such as Venice and Barcelona. The model highlights key components for effective management, as the role that restaurants and taverns have in keeping the informal network of small stores and entrepreneurs together, collaborating and finding creative solutions to adapt and survive. These types of actors encourage historic and cultural tourism while simultaneously incorporating traditional and established commercial activities and inhabitants [98]. Therefore, public administrations and associations can use this social network analysis to identify relevant actors that help to remedy or minimize gentrification processes, as in the Berlin district of Kreuzberg, Gracia in Barcelona, Palermo in Buenos Aires, and Le Marais in Paris.

Secondly, the study shows that urban tourism entrepreneurs should not try to undertake their ventures in solitude. Entrepreneurship is a social activity, as our social network analysis highlights. For the success of new urban businesses, it is critical to participate in activities that promote their networking [93], such as joining associations or collaborating in commercial activities.

### 4.3. Limitations and Future Research

This article presents an in-depth, carefully selected case study, although the model has only been tested in a unique case. Therefore, further research should include other cases, to compare and validate our model in various settings and it would be probably more interesting to see a comparative study instead of a single case study, for example a comparative study between Barrio de Las Letras in Madrid and Barrio Italia in Santiago de Chile, or Palermo or San Telmo in Buenos Aires.

The study also focuses on exploring how important it is to educate entrepreneurs in how to generate historic and cultural tourism ecosystems, given that they require specialized commerce and services, complex collaboration, and sophisticated foreign tourism demands. Additional research might incorporate a process vision [40] to explore the evolution or changes in the creation, development, and maintenance of cultural and historic tourism ecosystems. Finally, given the particularities of the case study, a place where not only do we find the houses of famous writers such as Cervantes or Lope de Vega, but also the houses where important literary works were created, we believe that an interesting avenue for future research is literary tourism, which has grown into a commercially significant phenomenon [77] even though literary tourism remains under researched in the academic world.

To conclude, this work differs from previous studies for several reasons. First, this study integrated key stakeholders, not just commercial stores, and entrepreneurs but also cultural institutions, such as museums and scientific, literary, and artistic centers. Secondly, we have carefully analyzed the interaction between commercial and other types of productive activities in urban centers. Social relationships and collaborations among small retailers and various entrepreneurs located in historic city centers have rarely been analyzed in prior literature. Yet the collaboration of cultural institutions, the hospitality industry, and retail can promote cultural experiences and historical tourism [41]. Thirdly, the study explicitly denoted the role of management mechanisms, such as the small business association, its manager, and the informal network of entrepreneurs and merchants, which has been scarcely addressed in previous research.

**Author Contributions:** Conceptualization, B.G.H. and E.S.; methodology, B.G.H. and E.S.; software, B.G.H., E.S., and P.C.-V.; validation, B.G.H., E.S., and P.C.-V.; formal analysis, B.G.H., E.S., and P.C.-V.; investigation, B.G.H., E.S., and P.C.-V.; resources, B.G.H., E.S., and P.C.-V.; data curation, B.G.H., E.S., and P.C.-V.; writing—original draft preparation, B.G.H. and E.S.; writing—review and editing, B.G.H. and P.C.-V.; visualization, B.G.H., E.S., and P.C.-V.; supervision, B.G.H., E.S., and P.C.-V.; project administration, B.G.H., E.S., and P.C.-V.; funding acquisition, B.G.H. and E.S. All authors have read and agreed to the published version of the manuscript.

**Funding:** This research was funded by Facultad de Economía y Negocios. Universidad del Desarrollo, Santiago de Chile (Chile) and Asociación de Comerciantes Barrio de Las Letras de Madrid.

**Acknowledgments:** We would also like to thank Asociación de Comerciantes Barrio de Las Letras de Madrid, Universidad de Alcalá (Spain), Universidad del Desarrollo (Chile) and Dirección General de Comercio y Hostelería del Ayuntamiento de Madrid for his administrative and technical support.

**Conflicts of Interest:** The authors declare no conflict of interest.

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
