# Peer review of "A Sustainable Management Model for Cultural Creative Tourism Ecosystems"

_sustainability, doi:10.3390/su12229554_

Round 1

Reviewer 1 Report

Congratulations, very interesting and well conducted research. It would be probably more interesting to see a comparative study instead of a single case study, but still the input into our contemporary knowledge is more than satisfactory.

Author Response

Thank you very much for your comments and suggestions to improve our document.

We include information about two possible comparative studies between Barrio de Las Letras in Madrid and Barrio Italia in Santiago de Chile or Palermo or San Telmo in Buenos Aires (lines 513-15).

We are working in a future paper to compare the Management Model in two cities (Barrio Italia in Santiago de Chile and Barrio de Las Letras in Madrid) and we are planning a new study to compare some of the tourist, creative and cultural neighbourhoods in Madrid (Las Salesas, Lavapiés, Conde Duque and Malasaña districts), so we are very grateful for your contribution.

Reviewer 2 Report

Dear authors,

Congratulations for your paper. It is very interesting and reflects some knowledge about the related area in Madrid.

It could be nice to add a map of Madrid in order to show where this Barrio de las Letras is located. With this the readers could understand a bite better the importance of this area.

Please review the english in order to improve the text.

Check all the references, looking for the information provided by the journal.

Author Response

Thank you very much for your comments and suggestions to improve our document. 

As the reviewer propose, we add a map of Madrid tourist areas in order to show where this Barrio de las Letras is located to present the importance of this area. (we include a new Figure 3 in line 256) and we change all the references numbers and figures located after Figure 3. In this sense, we included the new reference of the Map of Tourist Areas (Source: esmadrid.com [72]).

Additionally, English has been improved by a native English speaking.

And finally, we check all the references, looking for the information provided by the journal.